

# Distinctive personality profiles of fibromyalgia and chronic fatigue syndrome patients

Jacob N. Ablin[1,2], Ada H. Zohar[3], Reut Zaraya-Blum[3] and
Dan Buskila[4,5]

[1] Institute of Rheumatology, Tel Aviv Sourasky Medical Center, Tel Aviv, Israel
[2] Sackler School of Medicine, Tel Aviv University, Tel Aviv, Israel
[3] Department of Clinical Psychology, Ruppin Academic Center, Israel
[4] Department of Medicine H, Soroka Medical Center, Beer Sheva, Israel
[5] Faculty of Health Sciences, Ben-Gurion University of the Negev, Beer Sheva, Israel

## ABSTRACT

**Objective:** The current study is an innovative exploratory investigation, aiming at identifying differences in personality profiles within Fibromyalgia Syndrome (FMS) and Chronic Fatigue Syndrome (CFS) patients.

**Method:** In total, 344 participants (309 female, 35 male) reported suffering from FMS and/or CFS and consented to participate in the study. Participants were recruited at an Israeli FM/CFS patient meeting held in May 2013, and through an announcement posted on several social networks. Participants were asked to complete a research questionnaire, which included FMS criteria and severity scales, and measures of personality, emotional functioning, positivity, social support and subjective assessment of general health. In total, 204 participants completed the research questionnaire (40.7% attrition rate).

**Results:** A cluster analysis produced two distinct clusters, which differed significantly on psychological variables, but did not differ on demographic variables or illness severity. As compared to cluster number 2 (N = 107), participants classified into cluster number 1 (N = 97) showed a less adaptive pattern, with higher levels of Harm Avoidance and Alexithymia; higher prevalence of Type D personality; and lower levels of Persistence (PS), Reward dependence (RD), Cooperation, Self-directedness (SD), social support and positivity.

**Conclusion:** The significant pattern of results indicates at least two distinct personality profiles of FM and CFS patients. Findings from this research may help improve the evaluation and treatment of FM and CFS patients, based on each patient's unique needs, psychological resources and weaknesses, as proposed by the current trend of personalized medicine.

## INTRODUCTION

Fibromyalgia (FMS) is a syndrome characterized by chronic widespread musculoskeletal pain and tenderness, associated with disturbed sleep patterns, chronic fatigue and a

Corresponding author
Jacob N. Ablin, Jacobab@tlvmc.gov.il

spectrum of additional functional symptoms. While the pathogenesis and etiology of FMS remain incompletely understood, a leading paradigm in this aspect currently holds that FMS is the result of a process of pain centralization, whereby the central nervous system has become extremely hyper-sensitive to the processing and transmission of pain, thus causing amplification of painful as well as non-painful stimuli and leading to a state of chronic pain (*Clauw, 2015*). Chronic Fatigue Syndrome (CFS) is a clinical syndrome characterized by fatigue lasting over six months which is not associated with physical effort and is not relieved by rest (*Afari & Buchwald, 2003*). Considerable clinical overlap existed between FMS and CFS, with many patients fulfilling criteria for both conditions simultaneously. Similar to other chronic medical conditions, psychiatric comorbidity such as anxiety and depression have been documented in a portion of FMS patients, while other patients appear to be surprisingly psychologically resilient in the face of ongoing pain and fatigue (*Giesecke et al., 2003*).

The purpose of the current study was to evaluate personality types found among FMS/CFS patients. The study was not specifically designed to compare FMS patients to CFS patients, but rather to characterize this generally overlapping population of patients in terms of personality types.

In the current study, we have focused on the following four aspects of personality and psychological coping: Alexithymia—the inability to identify and describe emotions in the self; Type D personality—the tendency towards negative affectivity (NA) and social inhibition (SI) and personality components, based on the psycho-biological model of Cloninger regarding temperament and character; Level of positivity—self-confidence, optimism and satisfaction with life; Social support.

Alexithymia has been studied in patients suffering from chronic pain (*Celikel & Saatcioglu, 2006*; *Lumley, Smith & Longo, 2002*) and FMS (*Sayar, Gulec & Topbas, 2004*; *Steinweg, Dallas & Rea, 2011*). In a recent study, *Castelli et al. (2012)* reported alexithymia traits in 20% of a sample of FMS patients. Thus, it has previously been suggested that the inability to correctly identify physical manifestations of emotions makes alexithymic individuals susceptible to incorrectly attributing innocent physical symptoms to physical disease (*Tuzer et al., 2011*). Originally identified by *Denollet et al. (1996)* as a predictor of long-term mortality among patients suffering from coronary heart disease, type D personality was characterized by a tendency towards NA together with SI. Type D personality is strongly associated with both musculoskeletal pain, psychosomatic symptoms (*Condén et al., 2013*) and sleep disorders (*Condén, Ekselius & Åslund, 2013*).

The psycho-biological model of temperament and character (*Cloninger, Svrakic & Przybeck, 1993*) suggests the existence of characteristic temperament dimensions, defined in terms of individual differences in associative learning in response to novelty, danger or punishment and reward. Thus, Novelty seeking (NS) is described as a bias towards behaviors such as exploratory activity in response to novelty, impulsive decision making, as well as quick loss of temper in response to frustration. Harm avoidance (HA) biases individuals to habitual pessimistic worry, fear of uncertainty, shyness and rapid fatigability. Reward dependence (RD) is a tendency to sentimentality, to social attachment and dependence on the approval of others. Persistence (PS) refers to a tendency to

continue specific behavior despite feelings of frustration, fatigue or lack of reward (*Cloninger & Svrakic, 1997*). According to the "unified biosocial theory of personality," NS is associated with low basal dopaminergic activity, HA with high serotonergic activity, and RD with low basal noradrenergic activity (*Cloninger, 1986*). The psycho-biological model also identifies three dimensions of character, namely self-directedness (SD), Cooperativeness (CO) and Self-transcendence (ST). Genetic analysis has confirmed a hereditability component between 50 and 65% in each of the personality traits (*Heath, Cloninger & Martin, 1994*). *Cohen et al. (2002)* investigated the association between FMS and the serotonin transporter promoter region polymorphism, and the relationship to anxiety-related personality traits in FMS patients. A subsequent study investigated the association between FMS and the dopamine D4 receptor gene and the relationship to NS trait (*Dan et al., 2004*). *Glazer et al. (2010)* comparing the personality of FMS patients and their relatives with and without FMS found that FMS patients, as well as their relatives with FMS, had higher scores on HA than relatives without FMS. Despite these studies, a recent systematic review failed to identify a specific "FMS personality;" instead it was proposed that personality may act as a filter, that modulates a person's response to psychological stressors and that certain personalities may facilitate translation of these stressors to physiological responses which culminate in FMS (*Malin & Littlejohn, 2012*). While various aspects of both personality as well as psychiatric comorbidity have been frequently studied in the FMS syndrome, less attention has been focused on aspects of resilience and positivity, which is defined as a general dispositional determinant of subjective wellbeing, which may account for individual variation and stability in happiness, despite environmental challenge (*Kozma, Stone & Stones, 2000*).

Social support, the perception of the individual that he is cared for and loved, esteemed, and a member of a network of mutual obligations (*Cobb, 1976*) may have a protective effect versus a broad range of pathological conditions. Social support is directly associated with the severity of physiological and psychological symptoms and may moderate the health-related effects of stress (*Procidano & Heller, 1983*; *Sarason et al., 1985*). Patients suffering from FMS and CFS have previously been reported to suffer from low levels of social support (*Schoofs et al., 2004*).

## METHODS

### Participants

Individuals who self-identified as suffering from FMS or CFS, female or male were recruited through an appeal made at an FMS patient organization meeting held in Tel Aviv, Israel on May 2013, as well as through internet forums of FMS/CFS patients. No reward was offered, for participation. Participants had the option of being notified of the study results. The study was approved by the IRB of the Ruppin Academic Center (IRB reference number 8/2013).

### Procedure

After providing consent to participate in the study, participants were given access to an online self-report, using Qualtrics as the delivery system at http://www.qualtrics.com

website (Provo, UT, USA) (*Bryson, Turgeon & Choi, 2012*; *Passmore et al., 2002*). The online survey, which was anonymous, took about 20 min to complete, and participants were able to stop and restart as necessary, in order to minimize the discomfort FMS patients may experience during prolonged sitting. In total, 204 participants completed the entire survey while 140 (40.7%) completed it only partially.

## Tools and measures

### FMS

Meeting FMS diagnostic criteria as well as disease severity were determined through the Fibromyalgia Criteria and Severity Scales (FCSS) which are used for the diagnosis of FMS in epidemiological and clinical studies (*Wolfe et al., 2011*) and which are based on the 2010 proposed ACR criteria for the diagnosis of FMS (*Wolfe et al., 2010*). This questionnaire includes two scales: the widespread pain index (WPI) and the symptom severity scale (SSS). The results of these two scales are used both for establishing the diagnosis of FMS as well as for evaluating severity. Notably, in the current study a diagnosis of CFS was not specifically tested for separately; rather patients were surveyed for the fulfillment of FMS criteria alone as described. We adopted this approach due to the overwhelming clinical epidemiological overlap between FMS and CFS.

Alexithymia was assessed through the Toronto Alexithymia Scale (*Bagby, Parker & Taylor, 1994*) which measures inability to identify and describe emotions. The questionnaires includes 20 items on a Likert scale from 1 to 5 and measures three subscales: the Difficulty Describing Feelings subscale, the Difficulty Identifying Feeling subscale and the Externally-Oriented Thinking subscale. This tool has previously been translated into Hebrew and validated (*Zohar et al., 2011*).

## Personality evaluation

Personality was evaluated through the following questionnaires: Temperament and Character Inventory Revised (TCI-R) (*Cloninger, Przybeck & Svrakic, 1994*). In the current study, the shortened, 140 item version was used (TCI-140). This questionnaire includes seven scales: HA, RD, PS, SD, CO and ST. A validated Hebrew version was used in the current study (*Zohar & Cloninger, 2011*). Type D personality was assessed using the DS-14 questionnaire (*Denollet, 2005*) which includes seven items referring to NA and seven items referring to SI. Individuals who score 10 points or more on both NA and SI dimensions are classified as Type D personality. A validated Hebrew translation of this tool was utilized in the current study (31).

## Positivity

This construct was assessed by the Positivity Orientation Scale (P scale) (58). This tool is comprised of eight items assessing an individual's positive attitude about himself about his life and his attitude towards the future.

## RESULTS

In total, 344 participants (309 females and 35 males) participated.

Table 1 presents demographic details of the study participants.

**Table 1 Demographic data, ethnicity, level of religiosity, employment status and level of physical activity among study participants.**

| Age | 16–69 (mean: 41.74, SD: 12.09) | Frequency | % |
|---|---|---|---|
| Gender | Female | 309 | 89.8 |
| | Male | 35 | 10.2 |
| Marital status | Unmarried | 96 | 27.9 |
| | Married | 174 | 50.6 |
| | Divorced | 68 | 19.8 |
| | Widow | 6 | 1.7 |
| Educational level | Partial high school ($\leq$ 10 years) | 18 | 5.2 |
| | Full high school (12 years) | 102 | 29.7 |
| | $\geq$ 13 years | 84 | 24.4 |
| | First degree | 94 | 27.3 |
| | Second or third degree | 46 | 13.4 |
| Ethnicity | Jewish | 330 | 95.9 |
| | Arab–Muslim | 2 | 0.6 |
| | Arab–Christian | 1 | 0.3 |
| | Other | 11 | 3.2 |
| Religiosity level | Secular | 238 | 69.2 |
| | Traditional | 61 | 17.7 |
| | Orthodox | 38 | 11.1 |
| | "Haredi" (ultra-orthodox) | 6 | 1.7 |
| Employment status | Not working or studying | 115 | 33.4 |
| | Fully employed | 34 | 9.9 |
| | Partial employment | 100 | 29.1 |
| | Occasional work/volunteer | 95 | 27.6 |
| Level of physical activity | No physical activity | 51 | 14.8 |
| | Low physical activity | 151 | 43.9 |
| | Moderate activity | 114 | 33.1 |
| | High activity | 27 | 7.8 |
| | Very high activity | 1 | 0.3 |

## Patient characteristics

In total, 260 participants, which represented 75.6% of the total, met diagnostic criteria for FMS, while 84 participants (24.4%) did not. Notably, participants not meeting FMS criteria nonetheless suffered from chronic widespread pain, fatigue and other symptoms of variable severity and thus may be considered as representing sub-threshold FMS cases. In total, 235 (90.4%) of the participants who met FMS criteria were female. In total, 99 participants (28.8%) were characterized as type D personality, including 29.1% of females and 25.7% of males, respectively. No significant association was observed between gender and type D personality—$\chi^2(1) = 0.179$, p = 0.673 (NS) N = 344. The 30 percentage of participants meeting FMS criteria were characterized as type D personality compared with 25% of participants not fulfilling FMS criteria. No significant association was observed between type D and fulfilling FMS criteria $\chi^2(1) = 0.774$, p = 0.379 (NS), N = 344.

## Cluster analysis

In order to evaluate patterns of psychological coping, cluster analysis was performed on 204 participants whose questionnaires were complete. The analysis yielded two groups, which differed from each other on psychological variables including type D personality, alexithymia, positivity, social support, Cloninger's temperament and character domains. A total of 97 and 107 participants were classified into cluster 1 and cluster 2, respectively. Compared with cluster 2, cluster 1 was characterized by the following character dimensions: lower CO and lower SD. Cluster 1 was also characterized by the following temperament domains: lower PS, lower RD, and higher HA. This cluster was also characterized by higher levels of alexithymia, less social support, lower levels of positivity and higher frequency of type D personality. Table 2 presents mean values of psychological variables for each cluster.

## DEMOGRAPHIC CHARACTERISTICS OF CLUSTERS

After the process of cluster analysis, demographic parameters were compared between the two clusters including age, gender, educational level and employment status. No significant differences were found between cluster 1 and 2 regarding age, gender, marital status, educational level, and employment status. In order to test the hypothesis that the two clusters would not differ regarding FMS criteria, a Chi-Square Test for Independence was performed. The 79.4 percentage of individuals in cluster 1 and 72% of individuals in cluster 2 fulfilled FMS diagnostic criteria. In accordance with the hypothesis, no significant association was found between cluster designation and the diagnosis of FMS—$\chi^2(1)= 1.513$, $p = 0.216$ (NS), N = 204.

In order to compare the cluster analysis categories regarding the clinical parameters of FMS severity, as well as levels of physical activity, a Multivariate analysis of variance (MANOVA) was performed. In this analysis the independent variable was the cluster category and the dependent categories were the subjective health assessment, the extent of physical activity, the SSS, and the WPI. The results of the MANOVA indicated a significant difference between the cluster categories regarding the aggregate of dependent variables (Wilks' Lambda = 0.93, $F(4,199) = 3.56$, $p < 0.01$). Univariate analysis demonstrated a significant difference between the cluster categories regarding the SSS—$p < 0.0001$, with symptom severity significantly higher among individuals in cluster 1 (Mean = 9.11, SD = 2.18) compared with cluster 2 (Mean = 7.99, SD = 2.19). No significant difference was found when comparing cluster 1 with cluster 2 regarding subjective health assessment, WPI and levels of physical activity.

## DISCUSSION

The current study constitutes a novel exploratory approach towards identifying psychological patterns and personality aspects of resilience among patients suffering from FMS and CFS. In this study, we have attempted to elucidate the ways in which personality patterns interact with styles of coping with illness. While we recruited patients who self-reported a diagnosis of FMS, a significant proportion of the participants were found to be sub-threshold regarding the diagnosis of FMS, according to current diagnostic criteria.

**Table 2 Cluster analysis presenting mean values of psychological variables per cluster group.**

|  | Cluster 1 (N = 98) Mean (SD) | Cluster 2 (N = 106) Mean (SD) |
|---|---|---|
| Self-transcendence (ST) | 44.73 (12.02) | 44.49 (12.65) |
| Cooperativeness (CO) | 74.46 (8.36) | 82.60 (7.27) |
| Self-directedness (SD) | 57.59 (9.09) | 73.79 (7.54) |
| Persistence (PS) | 58.81 (10.67) | 69.69 (9.16) |
| Reward dependence (RD) | 65.53 (9.99) | 71.22 (8.38) |
| Harm avoidance (HA) | 73.76 (9.69) | 59.73 (11.07) |
| Novelty seeking (NS) | 55.87 (7.97) | 54.23 (8.14) |
| Alexithymia | 46.07 (10.93) | 34.75 (7.19) |
| Social support | 43.65 (11.52) | 59.00 (9.21) |
| Positivity | 2.79 (0.65) | 3.62 (0.51) |
| Type D personality | 0.755 (0.43) | 0.20 (0.04) |

In the analysis of results we have chosen to compare this group of individuals, who suffer from varying degrees of chronic pain and fatigue, with individuals who fulfil the FMS diagnostic criteria. Individuals fulfilling the diagnostic criteria were found to rate significantly lower on scales of social support, positivity, physical activity, and subjective health assessment. These groups also differed in the level of SD, which was lower among the criteria-positive individuals compared with the sub-threshold individuals. This finding is in accordance with previous research, which indicates that the SD of FMS patients is lower than that found among healthy controls (*Gencay-Can & Can, 2012*). SD is a trait which indicates the extent to which an individual can depend on himself and on his capabilities, allowing him to feel responsible for his own fate, resourceful and hopeful (*Cloninger, Svrakic & Svrakic, 1997*). Low levels of SD (as well as low levels of CO and high levels of HA) have been described in chronic pain patients (*Conrad et al., 2007*).

The most significant findings of the current study are related to the classification of the study participants (both FMS-criteria positive and sub-threshold individuals) into two clusters, based on the psychological styles of coping identified. This analysis yielded two clusters which clearly differ in their psychological profile, while not being significantly different on clinical grounds, and the clusters were nearly equal in the number of individuals they aggregated. The numerically larger cluster (107 vs. 97) was highly resilient in their psychological profile. Individuals in this cluster had a frequency of type D personality which was similar to the general population.

They were as high as healthy controls in the character traits of SD and CO, and in the temperament trait of PS, and as low in the temperament trait of HA (*Cloninger & Zohar, 2011*). This result is very similar to that of *Leombruni et al. (2016)* who also found two distinct personality clusters for FM patients, with the bigger cluster characterized by resilient personality profile and emotional style. The first group of patients was the mal-adapted cluster, characterized by higher levels of HA and alexithymia, higher frequency of type D, and lower levels of CO, SD, PS, RD positivity and social support. No significant differences were identified between the groups regarding the traits of ST and NS. The clear

clustering of patients between these two groups and the striking differences found between them, indicate the existence of two uniquely separate styles of coping among FMS patients and sheds doubt on the findings of previous studies which implied the existence of one homogeneous personality pattern among FMS patients (*Anderberg et al., 1999*; *Gencay-Can & Can, 2012*) and other patients suffering from chronic pain (*Conrad et al., 2013*). The emotional profile which emerges from the characteristics of the first group indicates a generally less adaptive pattern, associated with a decreased well-being. Due to high levels of HA, these individuals are more prone to be cautious, nervous, passive, negativistic, insecure and pessimistic. High levels of carefulness and pessimism may be associated with difficulty with expressing emotions, due to fear of negative implications (*Cloninger & Zohar, 2011*). Individuals with high levels of alexithymia, as in this group, experience difficulty in identifying their own emotions and in differentiating between these emotions and physical sensations which are associated with emotional stimulation. They may have difficulty articulating emotions and tend to be self-centered, inpatient and critical. The low levels of SD of such individuals is associated with low self-esteem, difficulty in taking responsibility, difficulty in setting long-term goals and in overcoming obstacles. Individuals in the first group are also characterized by low levels of PS and difficulty in coping with frustration. Low reward—dependence among these individuals has the advantage of less dependence on satisfying others and more independence; it also however carries the disadvantages of a tendency towards social withdrawal and isolation (*Cloninger, 2008*). Thus, individuals in this group are prone towards low levels of social support, and may not feel they have adequate bonds to fall back on (*Zimet et al., 1988*). Such individuals tend to have lower levels of positivity, lower self-esteem and higher degrees of pessimism (*Caprara et al., 2012*). Individuals in the first group were also found to have a higher frequency of type D personality, the characteristics of which appear to be in agreement with other personality characteristics identified among individuals in this group. Due to negative cognitive patterns, these individuals tend to experience difficulty in acquiring social support, tend to experience anxiety in social contexts and experience the surroundings as critical, thus augmenting negative feelings (*Denollet, 2005*).

Individuals in the second group, show a surprising psychological profile, of resilience and well-being, although they are no less prone to FM pain and fatigue. They are characterized by a healthier and more adaptive pattern of coping, compared with the first group. Due to lower levels of HA, these individuals tend to be freer of worries, energetic, extroverted and optimistic. The clear advantage of this pattern lies in the confidence such individuals feel when faced with danger or insecurity. On the other hand, such individuals are at an increased risk of exhibiting inadequate response to danger situations at the risk of actual harm. Individuals in the second group were also characterized by higher levels of RD and as such were more socially interactive and sensitive. They tend to be more tuned-in on social cues and to create more frequent relationships based on genuine affection and concern for others. They are frequently involved in efforts to satisfy others, as part of their social interactivity. These individuals were also characterized by higher levels of PS and setting higher personal goals. They increase their efforts in expectation of reward and view fatigue and frustration as challenges to be overcome. These individuals scored higher on

scales of SD, and thus tend to be more responsible, reliable and resourceful. They are more realistic in setting goals and striving towards them. They also show higher levels of empathy, patience, and compassion. All these qualities lead to higher levels of cooperation and social support. Individuals in this group showed lower levels of alexithymia, and a population prevalence of type D, leading to a more optimistic outlook and to less social withdrawal and negative feelings.

Our results indicate that the personality clusters identified did not differ regarding demographic characteristics including age, gender, marital status, levels of education, and employment. There was also no difference between the groups regarding the proportion of individuals who were FMS-criteria positive versus the sub-threshold individuals. Levels of widespread pain, as measured by the WPI index also did not differ between groups, indicating no difference in levels of chronic pain or pain distribution. The groups did however show a significant difference on the SSS which was significantly higher in the first group. Thus, individuals in the first group experienced higher levels of symptoms such as abdominal pain, non-refreshing sleep and cognitive difficulties.

The results of the current study have important potential therapeutic and research implications. Emotional profiles such as those identified in our study have previously been shown to be associated with various aspects of wellbeing and to be predictive of an individual's future health condition (*Josefsson et al., 2011*). Thus SD is strongly correlated to general mental health and absence of personality disorder (*Suchankova et al., 2011*). Higher levels of SD and CO have been found to be associated with better mental and physical health, increased social support, and improved coping with stressful situations (*Zohar et al., 2011*).

High levels of HA, on the other hand, have been associated with the behavioral pattern of fear and avoidance (*Conrad et al., 2007*) and are associated with a poor response to treatment in the context of chronic pain (*Asmundson, Norton & Norton, 1999*).

The group clustering described in the current study has the potential of leading to the development of patients—specific treatment plans, adjusted for the needs of FMS patients, in accordance with the style of coping identified. Thus, patients exhibiting high levels of alexithymia may benefit from interventions aimed at improving the ability to identify and express emotions, while patients exhibiting characteristics of the type D personality may gain through interventions aimed at improving social skills and reducing negative feelings. Targeted interventions aimed at strengthening personality characteristics which boost resilience, such as identified in our second group, may help patients not originally exhibiting these characteristics. Both cognitive treatment and anti-depressants have previously been shown to increase levels of SD, reduce vulnerability to depression and associated co-morbidities (*Bulik et al., 1998*; *Joyce, Mulder & Cloninger, 1994*). Empirical testing of interventions aimed at increasing SD and decreasing HA among FMS patients is a future challenge.

## Limitations

While the current study appears to have identified two distinct personality patterns among patients suffering from FMS and CFS, these patterns may in fact not represent a

dichotomous distinction, but rather a spectrum between which patients may be distributed. In addition, in the present study we have not looked at the effects of modifying factors such as anxiety and depression, which may well add important clinical information beyond the personality characteristics. Furthermore, in the current study we have not differentiated between patients suffering from FMS and those suffering from "pure" CFS (not fulfilling FMS criteria). Despite the large clinical overlap between these groups, they are not identical and future research may highlight personality-pattern differences between these two groups.

## Implications and conclusion

In the current study, we have identified two distinctive clusters of personality characteristics among patients previously diagnosed as suffering from FMS or CFS. These results draw attention to the heterogenic psychological characteristics of this patient population and to the necessity to avoid unwarranted generalization in characterizing these individuals. Additional research into the spectrum of personality characteristics of patients suffering from FMS/CFS, as well as further delineation of the differences between these groups, are called for in order to facilitate ideal personalized treatment of these individuals.

### Funding

The authors received no funding for this work.

### Competing Interests

Ada H. Zohar is an Academic Editor for PeerJ.

### Author Contributions

- Jacob N. Ablin conceived and designed the experiments, analyzed the data, wrote the paper, prepared figures and/or tables, reviewed drafts of the paper.
- Ada H. Zohar conceived and designed the experiments, analyzed the data.
- Reut Zaraya-Blum performed the experiments, reviewed drafts of the paper.
- Dan Buskila conceived and designed the experiments, reviewed drafts of the paper.

### Human Ethics

The following information was supplied relating to ethical approvals (i.e., approving body and any reference numbers):

Social and Community Sciences Institutional Review Board Approval: 8/2013.

### Ethics

The following information was supplied relating to ethical approvals (i.e., approving body and any reference numbers):

Ruppin Academic center social and community sciences Institutional review board, Number 8/2013.

## Data Deposition

The raw data has been supplied as Supplemental Dataset Files.

## Supplemental Information

Supplemental information for this article can be found online at http://dx.doi.org/10.7717/peerj.2421#supplemental-information.

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
