# Peer review of "Distinctive personality profiles of fibromyalgia and chronic fatigue syndrome patients"

_PeerJ, doi:10.7717/peerj.2421_

## Round 0.1 · original submission · Minor Revisions

Please edit the text to include the suggestions that have been made by the reviewers.

·

Basic reporting

First of all, Thank you for allowing me to read this article. The manuscript is well-written and conforms in both structure and content to the high standards of PeerJ.

Experimental design

The cross-sectional design is god enough to answer the research questions.

Validity of the findings

The findings and conclusions are warranted in light of the analyses conducted and the design at hand.

Additional comments

I have just a couple of minor changes that need to be done:
1. Please spell out FMS and CFS in the abstract and then its abbreviation. Do the same for CFS in the Introduction.
2.Please add sd for the means in Table 2.
3. If possible add a limitation and implication section in the discussion.

Reviewer 2 ·

Basic reporting

This submission didn't adhere to all PeerJ policies ; for references in the text, authors used Vancouver style and not the "Name. Year" style with an alphabetized reference list. Sometimes, the english was ambiguious for example lines 42-43 "exquisitely hyper-sensitive " but rather "extremely hyper-sensitive".

Experimental design

The objective of the study is not clear for me. I don't understand if the objective was to assess differences between patients with fibromyalgia and those with chronic fatigue syndrome, or was to assess differences between patients with fibromyalgia and chronic fatigue syndrome.
The assessment and the prevalence of CFS in the sample were not specified in the text.

Validity of the findings

no comments

Additional comments

The authors report an interesting study but the aim was not clear. We don't know if the objective was to assess differences between patients with FMS Versus those with CFS or to assess differences patients with FM and CFS.

Annotated reviews are not available for download in order to protect the identity of reviewers who chose to remain anonymous.

Reviewer 3 ·

Basic reporting

No comments

Experimental design

No comments

Validity of the findings

No comments

Additional comments

1. Were the online questionnaires completed anonymously?
2. Please indicate the body that gave ethical approval of the study. If ethical approval was not required (e.g. because participants only completed online questionnaires anonymously), please state so.

---

## Round 0.2 · accepted · Accept

All the reviewer's comments have been satisfactorily addressed.